# MULTI-AGENT TRAJECTORY PREDICTION WITH SCALABLE DIFFUSION TRANSFORMER

## ABSTRACT

Accurate prediction of multi-agent spatiotemporal systems is critical to various real-world applications, such as autonomous driving, sports, and multiplayer games. Unfortunately, modeling multi-agent trajectories is challenging due to its complicated, interactive, and multi-modal nature. Recently, diffusion models have achieved great success in modeling multi-modal distribution and trajectory generation, showing promising ability in resolving this problem. Motivated by this, in this paper, we propose a novel multi-agent trajectory prediction framework, dubbed Scalable Diffusion Transformer (SDT), which is naturally designed to learn the complicated distribution and implicit interactions among agents. We evaluate SDT on a set of real-world benchmark datasets and compare it with representative baseline methods, which demonstrates the state-of-the-art multi-agent trajectory prediction ability of SDT in terms of accuracy and diversity.

## 1 INTRODUCTION

Multi-agent spatiotemporal systems play a prevalent role in real-world scenarios and applications, such as autonomous driving, sports events, and other social interacting patterns. The motion of agents creates complicated behavior distribution on both individual and system levels, making it challenging to provide a precise and thorough understanding. Without additional prior knowledge, it is common to learn such multi-agent distribution from easy-to-collect trajectory information. The trajectory prediction model is required to capture both the multi-modality and interaction dynamics of agents.

There have been a number of existing works attempting to give a practical solution to multi-agent system modeling. The main optimization direction comes from two parts: multi-modal distribution learning and interaction relationship understanding. Popular related techniques include but are not limited to attention mechanisms (Kosaraju et al., 2019; Vemula et al., 2018), social pooling layer (Alahi et al., 2016), graph-based message transmitting (Guttenberg et al., 2016; Santoro et al., 2017), etc. Although the above works have made constant progress to achieve decent results, accurate prediction of general multi-agent systems is still far from being solved.

Inspired by recent successes of diffusion models in trajectory planning (Janner et al., 2022) and generation (Tevet et al., 2022), we propose scalable diffusion transformer (SDT), a novel trajectory prediction framework with diffusion process and modified transformer architecture to model multi-modal and multi-agent joint distribution naturally. Following the diffusion modeling procedure, the predictive trajectories are first sampled from Gaussian white noise and then go through the denoising process step by step to reconstruct the multi-agent dataset distribution. The model outputs denoising noise given environment context and diffusion intermediate noisy data. Since there is no specific constraint on context information and trajectories, SDT can be easily scaled to any general multi-agent systems to accomplish prediction or even generation tasks.

With the diffusion model, we are capable of matching high-dimensional multi-modality data without the use of Gaussian mixture model (GMM) or manual modality anchors (Varadarajan et al., 2022), which need explicit hyperparameter design or human prior knowledge. Besides, we have found that the diffusion model helps interaction relationship learning, as its process of gradual denoising and restoring the target distribution provides additional communication space for different agents.

We hope our scalable architecture and general problem formulation can give new insights into multi-agent trajectory prediction. Our main contributions can be summarized as follows:

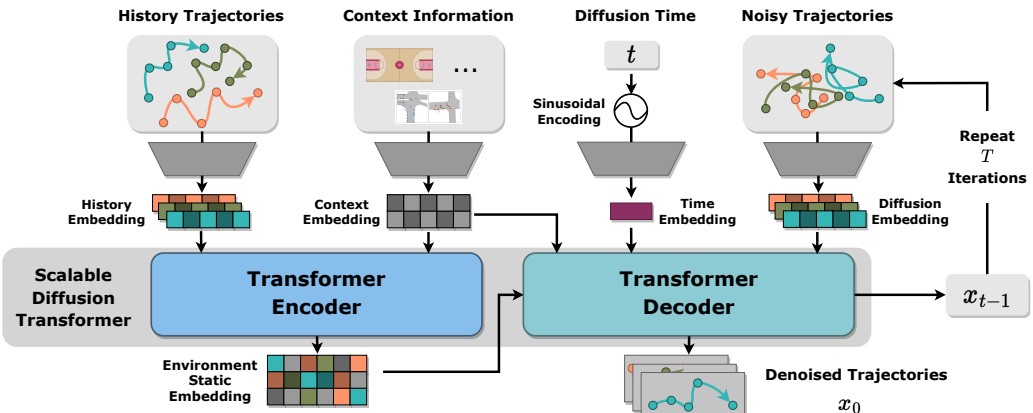

Figure 1: The overall framework of SDT. We first take history trajectories and context information as a static input to learn multi-agent environment representation through the encoder of SDT. Then following the DDPM, we take the environment representation, along with diffusion noise and timestep to decode the denoising noise of the next step. After $T$ iterations of the denoising process, we can obtain denoised trajectories $x_0$.

- We design a novel framework called scalable diffusion transformer (SDT) for multi-agent trajectory prediction by integrating the diffusion model into transformer architecture, which extends traditional transformer networks with better multi-modality modeling capability.

- We make novel and natural designs for transformer architecture to incorporate environment context and diffusion information and facilitate communication between agents. To our best knowledge, it is the first **general** multi-agent trajectory prediction framework with the diffusion model and can be easily scaled to different tasks such as generation.

- The experiment results show SDT achieves state-of-the-art performance in various domains of prediction tasks without further adaption or specific modification.

## 2 RELATED WORK

### 2.1 DIFFUSION MODEL

Diffusion model (Sohl-Dickstein et al., 2015; Song et al., 2020b) is a kind of generative model that has achieved great success in the text-to-image generation (Rombach et al., 2022). It takes Gaussian white noise as input and various text prompts as conditional guidance to output desirable images. Given its considerable potential in modeling complex and diverse distribution, recently many works have introduced it into the research of single-agent trajectory generation (Tseng et al., 2023) and decision-making (Wang et al., 2022). These methods encoded historical trajectory and environment return as conditions to generate qualified future trajectory. Here we extend this paradigm to multi-agent trajectory predictions and demonstrate its strong performance.

### 2.2 MULTI-AGENT TRAJECTORY PREDICTION

Predicting multi-agent future trajectories has been studied to model complex and dynamic interactions in many real-world scenarios such as traffic and sports. It is considered to be much more difficult than single-agent counterparts, as it requires accurate interaction modeling and rational reasoning based on single-agent prediction to represent a multi-agent system. Many learning-based methods were proposed, such as imitation learning (Sun et al., 2018), hidden Markov Models (Li et al., 2019; Zhan et al., 2018), deep graph networks (Li et al., 2020a), and transformer-based networks (Zhou et al., 2022). EvolveGraph (Li et al., 2020b) takes a latent graph to make explicit interaction modeling and proposes a dynamic mechanism to evolve the underlying interaction graph adaptively over time, while SceneTransformer (Ngiam et al., 2021) makes a simple but effective joint prediction of all agents with a unified architecture in a single forward pass. We follow the joint prediction idea and combine

the diffusion model with transformer architecture, taking advantage of both complex distribution matching and long-horizon sequential trajectory processing, respectively. MID Gu et al. (2022) and LED Mao et al. (2023) share the same idea with us but limit their usage to human motion prediction as they can't use contextual environment features. The most similar works to ours is MADiff (Zhu et al., 2023). It constructs a centralized-training-decentralized-execution (CTDE) decision-making framework in a multi-agent offline RL setting, which is different from our work as we focus on trajectory prediction and generation in a supervised learning manner with different neural network architecture designs.

## 3 PRELIMINARIES

### 3.1 PROBLEM SETTINGS

We consider a multi-agent spatiotemporal system with a fixed number of agents $N$. The trajectory of agent $\tau^i$ is a sequential array of states $\{s_t\}, t \in \{1, 2, \cdots, T\}$, where $s_t \in \mathbb{R}^D$ represents the state of an agent in $t$-th frame while $T$ is the number of frames. Our goal is to predict agent future trajectories $T_{1:F} = \{\tau^1_{1:F}, \tau^2_{1:F}, \cdots, \tau^N_{1:F}\}$ given its history trajectories $T_{-H:0}$ and environment context $C$, where $F$ and $H$ are future and history length of trajectories. The environment context may include time, map, scoreboard, and other information that is agnostic to the agents.

### 3.2 DIFFUSION FRAMEWORK

The diffusion process can be seen as a Markov noising and denoising process with $T_{1:F,t}$. Here we make simplification as $F_t = T_{1:F,t}$ referred to the future trajectory to avoid the confusion of trajectory timestep and diffusion timestep. The noising process can be written as:

$$q(F_t|F_{t-1}) = \mathcal{N}(\sqrt{\alpha_t}F_{t-1}, (1 - \alpha_t)\mathbf{I}), \tag{1}$$

where $\alpha_t \in (0, 1)$ are constant hyperparameters calculated by noising variance schedule. With small enough $\alpha_t$ and long enough nosing steps $T \to \infty$, we can approximate $F_T \sim \mathcal{N}(0, \mathbf{I})$. We use SDT as an approximation model to learn the reverse denoising procedure as:

$$p_\theta(F_{t-1}|F_t) = \mathcal{N}(F_{t-1}; \mu_\theta(F_t, t), \Sigma_\theta(F_t, t)). \tag{2}$$

The above equation can be simplified to learn the noise term $\epsilon \in \mathcal{N}(0, 1)$ using $q(F_t|F_0) = \sqrt{\bar{\alpha}_t}F_0 + \sqrt{1 - \bar{\alpha}_t}\epsilon, \bar{\alpha}_t = \prod_{i=0}^{t} \alpha_i$. The final optimization objective can be written as an MSE loss:

$$\mathcal{L} = \mathbb{E}_{F_0, t}[\|\epsilon - \epsilon_\theta(F_t, t)\|^2]. \tag{3}$$

The estimated mean of Gaussian $\mu_\theta(F_t, t) = \frac{1}{\sqrt{\alpha_t}}(F_t - \frac{1-\alpha_t}{\sqrt{1-\bar{\alpha}_t}}(\epsilon_\theta(F_t, t)))$ and the variance of Gaussian $\Sigma_\theta$ can be assigned with variance schedule.

## 4 METHODOLOGY

### 4.1 PIPELINE OVERVIEW

As illustrated in Fig.1, we construct a trajectory prediction pipeline based on the denoising diffusion probabilistic model (DDPM) (Ho et al., 2020), a popular diffusion model framework that provides both training and sampling algorithm. Following the diffusion model literature, the forward diffusion process maps original data to pure random samples by constantly adding small Gaussian noise with a variance schedule. For the denoising process, we propose the scalable diffusion transformer (SDT) to reverse the forward procedure and recover original data distribution with history trajectories and environment context as guidance prompts.

### 4.2 TRAJECTORY PREDICTION WITH DIFFUSION MODEL

The diffusion model can recover the distribution of the training dataset by sampling scheme where each time the model samples from the whole distribution randomly. However, this kind of randomness

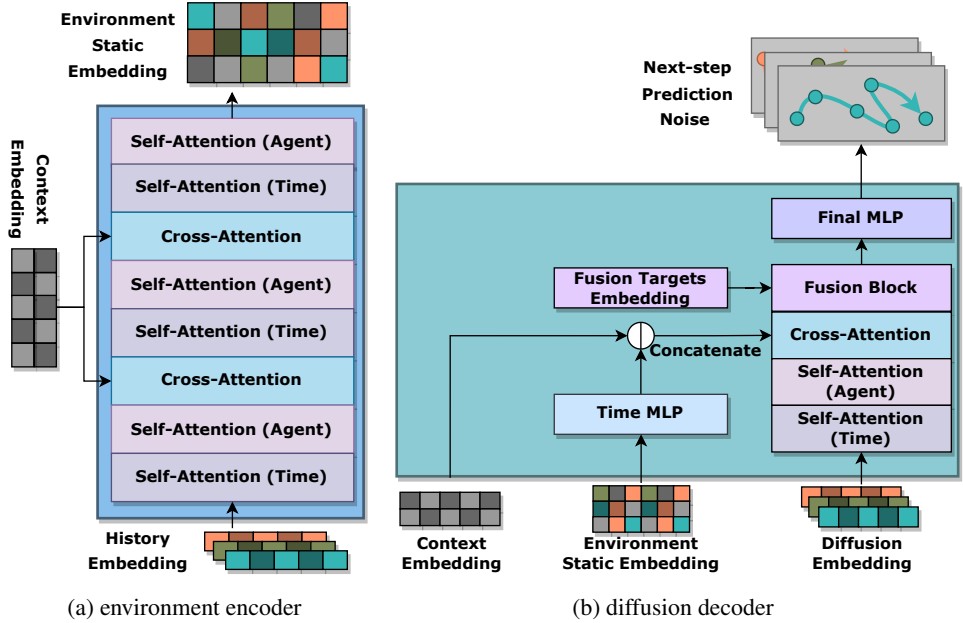

Figure 2: The structure of SDT. SDT is a typical encoder-decoder framework. The environment encoder learns context representation as guidance prompts, and the diffusion decoder predicts noise from learned guidance and diffusion information. The environment encoder consists of a list of self-attention and cross-attention blocks. The diffusion decoder takes diffusion noise as input tokens and processes embedding with self-attention blocks as well. We integrate learned environment embedding in the decoder by cross-attention mechanism and additionally apply a fusion block to bring diffusion timestep and target agents information to the model.

limits usage to obtain desirable outputs with certain features. A possible solution is the classifier-based guidance approach (Dhariwal & Nichol, 2021), which trains an explicit classifier to predict $p(y|F)$ given guidance information $y$ and use its gradients $\nabla_{F_t} \log p_\phi(y|F_t, t)$ to guide the diffusion sampling towards the conditional distribution $p(F_0|y)$. Besides, there is a simpler method called classifier-free guidance (Ho & Salimans, 2022; Ajay et al., 2022) which directly inserts $y$ as an additional input into the model and changes the diffusion procedure with the perturbed noise without performance loss as follows:

$$\epsilon' = \epsilon_\theta(F_t, t, \emptyset) + \omega(\epsilon_\theta(F_t, t, y) - \epsilon_\theta(F_t, t, \emptyset)) , \tag{4}$$

where $\omega$ is a scalar hyperparameter controlling the feature correlation with guidance $y$. In recent decision-making trajectory generation using offline reinforcement learning, the guidance $y$ is commonly set to the expected cumulative reward from the current environment step. We here take history trajectories $H$ and environment context $C$ as guidance.

With classifier-free guidance, the training loss objective can be written as

$$\mathcal{L} = \mathbb{E}_{F_0, t}[\|\epsilon - \epsilon_\theta(F_t, t, (1 - \beta)y + \beta\emptyset)\|^2] , \tag{5}$$

where $\beta$ is sampled from the Bernoulli distribution with probability parameter $p$. In practice, we set $p = 0$ if the environment features are not highly complex and found it doesn't hurt performance.

## 4.3 MODEL ARCHITECTURE

We propose a unified transformer architecture with an environment encoder and a trajectory decoder. The model predicts multi-agent trajectories in a centralized manner. The encoder extracts context embedding of the environment with a series of attention blocks, which works as high quality prompts indicating sequential and agent relationships of predicted future trajectories. The decoder makes a natural combination of environment embedding and diffusion noise with a specific fusion block to predict noise terms used for next-step denoising. We give detailed descriptions of the encoder and decoder in the following sections.

### 4.3.1 ENVIRONMENT ENCODER

The environment encoder, illustrated in Fig.2(a), aims to learn representations of the multi-agent system. To get a complete understanding of both the time dimension and agent dimension, we take advantage of a series of self-attention blocks, where each block contains two normalization layers, one multi-layer perception layer, and one multi-head attention layer, to fully study the correlations between different frames and agents. The environment context is integrated by cross-attention blocks, and is used to represent static agent-independent information in the system, like road graphs in autonomous driving or scoreboards, game clocks in NBA sports matches. With these attention blocks, we can obtain environment representation in the latent space with high quality.

### 4.3.2 TRAJECTORY DECODER

The learned representations of the environment can be seen as prompts to guide the decoder to output high-quality denoising noise of trajectories in each diffusion step. As seen from Fig.2(b), here we make diffusion noise as input tokens and combine the environment embedding by cross-attention mechanism. We design the fusion block to further inject diffusion timestep $t$ and required information of agents.

**Fusion Block.** Inspired by MotionDiffuse (Zhang et al., 2022), which uses a stylization block to bring timestep $t$ to the generation process, we here design a fusion block to strengthen the model with target agents embedding and inject diffusion timestep information in Fig.3. We first sum up diffusion timestep embedding after sinusoidal encoding $e_{time}$ and target embedding $e_{target}$. The target embedding is extracted from environment embedding with indexes that are pre-selected from the dataset. Here we only select a subset of indexes since only partial agents in the whole system are needed in making accurate predictions (e.g., in the traffic scenario we only pay attention to the target vehicle and its neighbors). Then we process the features through linear affine transformation with diffusion embedding X from the decoder and get output X′:

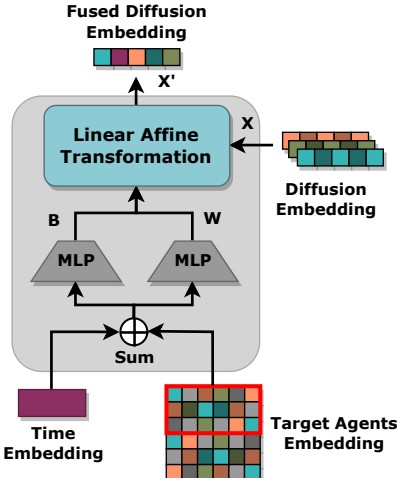

Figure 3: The fusion block. Fusion block fuses diffusion timestep and target agents information by linear affine transformation operation with diffusion embedding from the decoder.

$$B = \phi_B(e), W = \phi_W(e), X' = X \cdot W + B, \tag{6}$$

where $\phi_B, \phi_W$ is multi-perception layers and $(\cdot)$ denotes Hadamard product. By processing embedding through fusion block, the model is strengthened to identify diffusion timestep $t$ and required prediction of agents.

It is noticed that SDT is an index-free framework such that the order of agents can be arbitrary as long as the target indexes are correctly collected. Differing from traditional transformer networks which directly produce final prediction, the trajectory decoder outputs next-step diffusion noise in sampling progress each time. Given that we generate prediction in a centralized manner, the trajectory noise of agents can make communications through this step-by-step scheme to improve trajectory consistency. Although $T$ iterations inference will cause the time-consuming problem, it can be relieved by fast sampling methods like DDIM (Song et al., 2020a) or DPM-Solver (Lu et al., 2022).

## 5 EXPERIMENTS

We conduct experiments on two typical domains, autonomous driving and basketball games, to verify the effectiveness of our framework. We choose the argoverse2 (Wilson et al., 2023) as a representative dataset of autonomous driving and the widely used NBA dataset for basketball games. Each of them has different environment context and agent characteristics. We try to answer three questions: 1) Does SDT have better prediction performance compared to other general multi-agent trajectory prediction methods? 2) Does SDT have better multi-modality modeling capability and diversity? 3) How much does the diffusion process contribute to the prediction performance?

## 5.1 Environment Settings

**Argoverse2 motion forecasting dataset.** The argoverse2 dataset is a curated collection of 250,000 driving scenarios for training and validation. We follow the official configuration where each scenario is 11 seconds long and tracked agents are sampled at 10Hz. We take 5 seconds as history trajectory duration and 6 seconds for future prediction. Since each scenario has a target vehicle, we take its final history position and heading as references to build a scenario-centric coordinate. We choose the target vehicle and 7 nearest surrounding vehicles to be the prediction goals. To make reasonable guidance, we learned a static road graph as environment context similar to SceneTransformer.

**NBA sports dataset.** The dataset contains 2d trajectories of various basketball players recorded from 631 games in the NBA 2015-2016 season. In a basketball game, the players are divided into two teams to cooperate and compete, which means they make highly interactive behaviors and develop complex and dynamic relations. Following EvolveGraph's setting, we first downsampled the dataset to 2.5Hz and predicted the future 10 time steps (4.0s) based on the historical 5 time steps (2.0s). The guidance contains the history trajectory of the basketball and time clocks.

## 5.2 Baselines and Metrics

For the argoverse2 dataset, since we want to validate the advantage of SDT, we compare common LSTM baselines and our self-implemented version of SceneTransformer. To give a complete evaluation, we also make a single-agent comparison with competition leaderboard statistics. For the NBA dataset, we compared several common baselines, including graph-based and transformer-based methods.

We choose common distance-based metrics including average displacement error (ADE), final displacement error (FDE) and MR (miss rate). The ADE calculates the average L2 distance between predicted trajectories and ground truth and FDE calculates the final ones. Due to multi-modality in the multi-agent system, a single prediction cannot cover all possibilities. We further use $\text{minADE}_k$ and $\text{minFDE}_k$ to represent minimum ADE and FDE among $k$ predicted trajectories. In argoverse2 $k = 6$ while in the NBA dataset $k = 20$. MR measures the coverage of trajectories, calculating the percentage of trajectories where the prediction is within 2 meters of ground truth according to endpoint error. To further demonstrate the advantage of SDT in multi-modality modeling, we take additional two metrics from text-to-motion generation research: *Diversity* measures the variability in the resulting motion distribution, and *MultiModality* is the average variance given a single environment context.

## 5.3 Results

We report numerical results on both argoverse2 and NBA tasks. The prediction results calculate the average performance across three random seeds.

Table 1: Single-agent (marginal) trajectory prediction result on argoverse2 dataset. We choose typical methods with good performance to make comparisons. The methods from the leaderboard may take specific techniques to improve statistics while SDT works as a general method without additional modification. The result of SDT remains competitive.

| Methods | $\text{minADE}_6$ | $\text{minFDE}_6$ | MR |
|---|---|---|---|
| QCNet (Zhou et al., 2023) | 0.62 | 1.19 | 0.14 |
| TENET (Feng et al., 2023) | 0.70 | 1.38 | 0.19 |
| LaneGCN (Liang et al., 2020) | 0.91 | 1.96 | 0.30 |
| SceneTransformer (Ngiam et al., 2021) | 1.05 | 2.15 | 0.37 |
| SDT (Ours) | 0.94 | 1.93 | 0.29 |

**Results on Argoverse2 Dataset.** We evaluated our method on the validation set since additional metrics are introduced. Table 1 shows single-agent prediction results compared to methods from the leaderboard of argoverse2 motion prediction challenge. We choose typical methods with good performance to make a fair comparison. From the board SDT achieves medium-level performance which is better than general prediction methods like SceneTransformer, but worse than first-ranking

Table 2: Multi-agent trajectory prediction result on argoverse2 dataset. We here take SceneTransformer as the multi-agent baseline. The results demonstrate the accuracy and diversity of SDT on multi-agent system modeling.

| Methods | $minADE_6$ | $minFDE_6$ | MR | Diversity | MultiModality |
|---|---|---|---|---|---|
| SceneTransformer | 2.21 | 4.23 | 0.35 | 235.24 | 0.94 |
| SDT (Ours) | **1.84** | **3.75** | **0.33** | **286.27** | **1.35** |
| Real | - | - | - | 295.15 | - |

Table 3: Multi-agent trajectory prediction result on NBA sports dataset. We compare distance-based minADE and minFDE of 2.0s and 4.0s future among 20 prediction trajectories. Since *Diversity* and *MultiModality* are not evaluated by previous methods, we here only discuss these two metrics between our implemented methods and SDT. The best and the second-best results of each metric are marked as **bold** and underline, respectively.

| Methods | 2.0s | | 4.0s | | Diversity | MultiModality |
|---|---|---|---|---|---|---|
| | $minADE_{20}$ | $minFDE_{20}$ | $minADE_{20}$ | $minFDE_{20}$ | | |
| STGAT (Huang et al., 2019) | 0.91 | 1.39 | 2.47 | 3.86 | - | - |
| Social-STGCNN (Mohamed et al., 2020) | 0.90 | 1.43 | 2.35 | 3.71 | - | - |
| Social-Attention (Vemula et al., 2018) | 1.58 | 2.51 | 3.76 | 6.64 | - | - |
| Social-LSTM (Alahi et al., 2016) | 1.64 | 2.74 | 4.00 | 7.12 | - | - |
| Social-GAN (Gupta et al., 2018) | 1.52 | 2.45 | 3.60 | 3.60 | - | - |
| Trajectron++ (Salzmann et al., 2020) | 0.99 | 1.58 | 2.62 | 4.70 | - | - |
| NRI(dynamic) (Kipf et al., 2018) | 1.02 | 1.71 | 2.48 | 4.30 | - | - |
| EvolveGraph (Li et al., 2020b) | 0.74 | 1.10 | 1.83 | 3.16 | - | - |
| MADiff (Zhu et al., 2023) | 1.06 | 1.67 | 1.81 | 3.01 | **38.84** | **2.32** |
| SceneTransformer | 0.77 | 1.15 | 1.92 | 3.35 | 29.92 | 0.13 |
| SDT (Ours) | **0.69** | **1.05** | **1.79** | **2.88** | 35.98 | 0.17 |
| Real | - | - | - | - | 37.74 | - |

methods. It is reasonable as methods in competition universally use specific techniques to deal with traffic prediction problems, such as map feature engineering with prior knowledge and anchor-based guidance, while SDT merely make simple modifications on context information.

We mainly focus on multi-agent prediction performance, illustrated in Table 2. We have observed that SDT significantly outperforms SceneTransformer in accuracy metrics (minADE, minFDE and MR), which demonstrates the distributional modeling ability of the diffusion model. Furthermore, SDT achieves better multi-modality referred to *Diversity* and *MultiModality* performance. The diversity indicator is close to the real dataset with 3.0% error and better than SceneTransformer and multi-modality holds the upper hand as well.

**Results on NBA Sports Dataset.** Table 3 shows the prediction results on the NBA dataset. We can observe that SDT outperforms all other baselines on both minADE and minFDE. The baselines consist of different strategies to model dynamic relations and interactions between agents, such as graph-based representation, social pooling layers or attention mechanisms. Our proposed SDT enjoys both complex multi-modal distribution modeling ability from diffusion model and interaction understanding from sequential modeling using transformer, such that SDT achieves better performance which reduces the 4.0s minFDE by 4.3% with respect to the best baseline (MADiff). Aside from distance-based metrics, SDT have higher diversity and multi-modality than SceneTransformer. The diversity compared to that from the real dataset has only 7.3% error as well, which demonstrates that SDT is more suitable for multi-modal distribution modeling.

## 5.4 QUALITATIVE ANALYSIS

We plot qualified cases in Fig.4 to demonstrate that SDT produces high-fidelity, diverse, and consistent trajectories. From prior knowledge, we can easily know that the ball follows one of the players most time. Although we only provide the history of the ball and players, the predicted trajectories of players show a consistent correlation with future ground truth of the ball, demonstrating that SDT predicts plausible behaviors. We can also see reasonable cooperation and competition in figures. For example, in Fig.4(a), the predicted trajectory of players without the ball takes on the characteristics

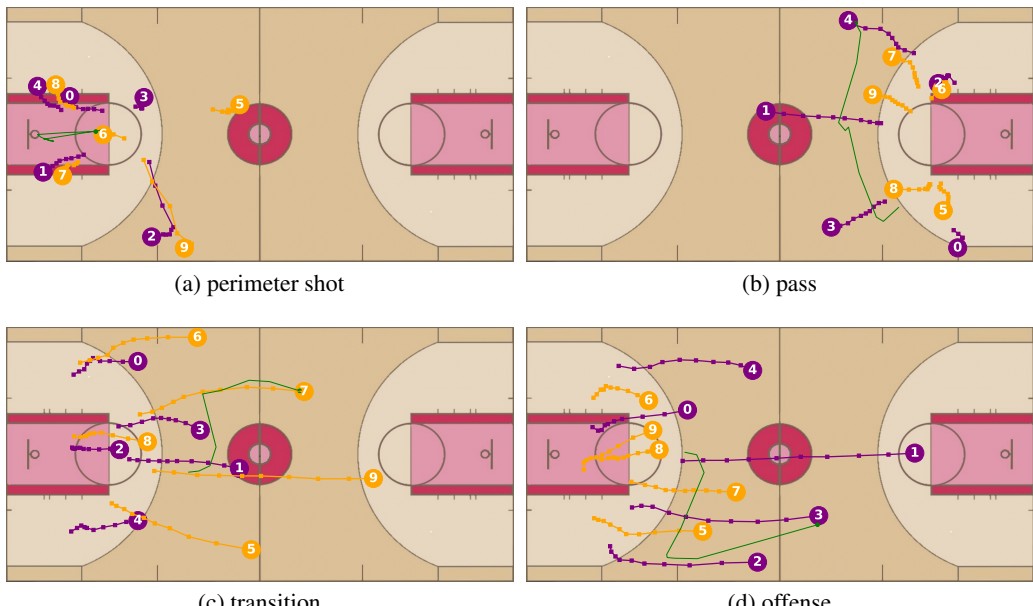

Figure 4: Qualitative analysis on NBA dataset. We visualize four examples from typical game periods. The numbered balls and lines with the same color indicate the team trajectories, and green lines are the trajectories of the basketball. We can see the trajectories show plausible behaviors with in-game strategies, achieving both accuracy and diversity.

of one-to-one defense, marking strong competition. In Fig.4(b) the ball passes through three players of the purple team cooperatively and in Fig.4(c) and Fig.4(d), The movement of players shows consistency at the tactical level and differences between individual positions. In general, SDT can understand and predict diverse behaviors with high accuracy.

## 5.5 ABLATION STUDY

The key insight of SDT is that the introduction of the diffusion model improves multi-modal distribution modeling ability and makes communication between agents more fully. We here conduct ablation experiments on NBA dataset to validate this idea. Three variants of SDT are designed: SDT-$k$ ($k = 30, 10, 4$) denotes that we use smaller $k$ diffusion timesteps instead of the original 50 steps to check the contribution of diffusion model in the framework; SDT-NT removes self-attention block of time dimension while SDT-NA removes self-attention block of agent dimension in the decoder to check how much can the iterative communication influence the performance.

As is observed in Fig.5, the performance of SDT decreases as diffusion timesteps become smaller, and a large decline amplitude appears in $k : 50 \rightarrow 30$. This may indicate that a complete diffusion process plays a significant role in expressiveness and distribution modeling, and an incomplete process may even be worse than 'no' process since SDT-4 outperforms SDT-10 and SDT-30. Furthermore, by comparing SDT with SDT-NT and SDT-NA, we can obviously conclude that iterative communication in the diffusion process helps the network to obtain a better understanding of the interaction between agents and time-sequential relationship among single trajectories.

## 6 GENERATION APPLICATION

In the field of autonomous driving, the driving data collected from the real world follows a long-tail distribution that most of the data for ordinary scenarios doesn't contain much value in training, and only a small amount of data shows interactive dynamics. This kind of data imbalance would hurt the performance of the downstream planning model, causing success rate loss in both general scenarios and complex corner cases. Since prediction and generation both require a comprehensive

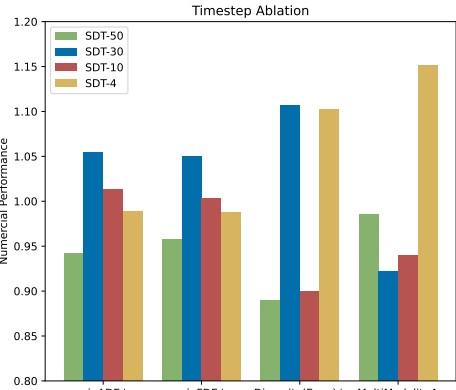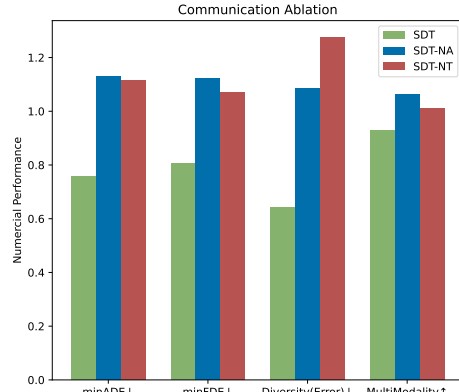

Figure 5: Ablation study. Here we use the diversity error between the real dataset and the model as the metric. The left figure shows the result of decreasing diffusion step (i.e. weakening the contribution of the diffusion model), where we can see a complete diffusion process is beneficial for both accuracy and diversity. The right figure demonstrates that iterative communication on both agent and time dimensions is significant to achieve high performance.

understanding of the multi-agent system and can be trained in a supervised learning manner, we can easily utilize SDT to perform trajectory generation without much modification.

We have conducted experiments on traffic generation with an internal dataset in order to solve the problem of data imbalance. To reproduce traffic segments with certain interactive traffic events (lane change, cut-in, overtake, etc.), we learned the representation of these events as additional guidance prompts and added a cross-attention block to the diffusion decoder to bring it in. With the map and history trajectory unchanged, we can generate different interactive scenarios by changing the event guidance, which still conforms to human driving distribution. We then combined the generated data with the original data to make a balanced dataset. The experimental results show that the planning model trained by the balanced dataset has a decent improvement in the pass rate of complex corner cases without losing performance in the general scenarios.

## 7 CONCLUSION

We have presented SDT, a general multi-agent trajectory prediction framework that makes use of history trajectories and environment context information. SDT is a classifier-free diffusion model featuring encoder-decoder transformer architecture to predict next-step noise in a centralized manner. This combination improves both diversity and accuracy with better multi-modal modeling. The experiments demonstrate that SDT has superiority in multi-agent trajectory prediction and can be easily transferred to similar tasks such as conditional trajectory generation.

It is worth noticing that SDT has a limitation that it is difficult to perform modality discrimination, which is caused by the random sampling nature of diffusion models. The output results of the same modality increase the sampling times of the prediction process. We would be interested in a better theory of modality control in the research of diffusion models and take it as future work.

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
