# OpenReview forum: "Multi-agent Trajectory Prediction with Scalable Diffusion Transformer"
_ICLR.cc/2024/Conference — ICLR 2024 Conference Withdrawn Submission_

### Official Review · Reviewer_VrGo · 2023-10-29

**Soundness:** 2 fair
**Presentation:** 2 fair
**Contribution:** 2 fair
**Rating:** 3
**Confidence:** 3

**Summary:**

This paper proposed a multi-agent trajectory prediction model based on diffusion models, named scalable diffusion transformer (SDT). The proposed SDT model consists of 1) a transformer encoder that encodes environment embeddings from historical trajectories and context information, and 2) a transformer decoder that iteratively diffuses the predicted joint trajectories conditioned on the environment static embedding. SDT was tested on the Argoverse 2 dataset for vehicle trajectory prediction and on the NBA dataset for human trajectory prediction. Especially, it achieved state-of-the-art results on the NBA dataset.

**Strengths:**

1. Applying the diffusion model to multi-agent trajectory prediction is a sensible and promising direction.
2. The proposed SDT model achieved state-of-the-art results on the NBA dataset.

**Weaknesses:**

Even though I found the idea of using diffusion models for multi-agent trajectory prediction interesting and sensible, the paper lacks novelty and sufficient technical contributions to advance the research on this topic.

1. The paper does not place itself well among the latest literature on multi-agent trajectory prediction in autonomous driving. In particular, the authors should compare the proposed SDT model against MotionDiffuser [1], a diffusion-based joint trajectory prediction model that achieves state-of-the-art performance on WOMD. In my opinion, SDT is a fairly straightforward adoption of diffusion for multi-agent trajectory prediction. Compared to MotionDiffuser, this work lacks an in-depth analysis of the model design (e.g., latent space in MotionDiffuser) and does not sufficiently show the advantages of diffusion models in the multi-agent trajectory prediction problem (e.g., controllable trajectory synthesis in MotionDiffuser). Also, the authors should compare SDT with other joint trajectory prediction models that do not rely on diffusion, for example, JFP [2] and QCNetXt [3], and discuss the advantages of diffusion compared to the other methods.

2. The performance of SDT on the argoverse2 dataset is not very satisfying. Also, the results are not well presented and diagnosed. In the marginal trajectory prediction task. the proposed SDT model performs much worse than the current state-of-the-art models. It performs on par with LaneGCN (Liang et al. 2020), which is a fairly outdated model. In the multi-agent trajectory prediction task, SDT is only compared against SceneTransformer. SDT's prediction accuracy is, in fact, quite far away from the state-of-the-art results from the leaderboard. For example, QCNetXt achieved a $minFDE_6$ of 1.02, while SDT's $minFDE_6$ is 3.75. While the authors' explanation is reasonable (i.e., those SOTA models have adopted sophisticated techniques to best tailor their models for the traffic prediction problem), it is misleading to not show a comprehensive comparison of SDT against SOTA results from the leaderboard, especially for the multi-agent trajectory prediction task. Also, while I don't think achieving SOTA performance is the only way to show the merits of a prediction model, the current results on Argoverse 2 are unfortunately not sufficient to demonstrate the advantages of using diffusion for traffic prediction. Why should we prefer diffusion over other methods for traffic prediction, if its performance is far away from SOTA, especially considering the significant computational cost of diffusion during the inference time?

3. The results of traffic generation could potentially strengthen the motivation behind adopting the proposed SDT model. However, the generation experiments were only conducted on an internal dataset and the results were not disclosed. The authors' description of the experimental results cannot be trusted and is not convincing, without solid and detailed numerical results. It would also be better to conduct additional experiments on public datasets so that the results can be reproduced.

[1] Jiang, Chiyu, et al. "MotionDiffuser: Controllable Multi-Agent Motion Prediction using Diffusion." Proceedings of the IEEE/CVF Conference on Computer Vision and Pattern Recognition. 2023.\
[2] Luo, Wenjie, et al. "Jfp: Joint future prediction with interactive multi-agent modeling for autonomous driving." Conference on Robot Learning. PMLR, 2023.\
[3] Zhou, Zikang, et al. "QCNeXt: A Next-Generation Framework For Joint Multi-Agent Trajectory Prediction." arXiv preprint arXiv:2306.10508, 2023.

**Questions:**

Why is the proposed model featured as scalable? Which parts of the model design make it particularly scalable than other diffusion models or multi-agent trajectory prediction models?

---

### Official Review · Reviewer_xv4S · 2023-10-30

**Soundness:** 3 good
**Presentation:** 2 fair
**Contribution:** 1 poor
**Rating:** 3
**Confidence:** 5

**Summary:**

This work proposes Scalable Diffusion Transformer, a denoising diffusion model for multi-agent trajectory prediction employing a Transformer architecture for agent interactions and temporal modelling.
This work follows the initial formulation set out in DDPM with the base architecture being a Transformer.
Experiments are performed on the Argoverse 2 vehicle motion forecasting dataset as well as the NBA dataset.
An ablation study is presented to show the importance of the number of diffusion timesteps and the importance of both social and temporal attention layers.

**Strengths:**

- I thought the motivation of the problem was quite compelling, with multi-modality being nicely addressed with diffusion models, as opposed to the classic GMMs.

- The architecture figures throughout section 4 are quite clear and well presented.

- The fusion block is an interesting part of the architecture. However, it is unclear how crucial this is to the functioning of the architecture.

**Weaknesses:**

- Novelty: This work claims to be the first to perform multi-agent motion prediction using diffusion models, but this is simply not true. [1,2,3] are all recent methods that were not cited, and are quite similar to the proposed approach. Of particular interest is MotionDiffuser which also uses attention layers to model the multi-agent diffusion problem. I believe this needs to be addressed.

[1] Xu, Chejian, Ding Zhao, Alberto Sangiovanni-Vincentelli, and Bo Li. "DiffScene: Diffusion-Based Safety-Critical Scenario Generation for Autonomous Vehicles." In The Second Workshop on New Frontiers in Adversarial Machine Learning. 2023.

[2] Jiang, Chiyu, Andre Cornman, Cheolho Park, Benjamin Sapp, Yin Zhou, and Dragomir Anguelov. "MotionDiffuser: Controllable Multi-Agent Motion Prediction using Diffusion." In Proceedings of the IEEE/CVF Conference on Computer Vision and Pattern Recognition, pp. 9644-9653. 2023.

[3] Zhong, Ziyuan, Davis Rempe, Danfei Xu, Yuxiao Chen, Sushant Veer, Tong Che, Baishakhi Ray, and Marco Pavone. "Guided conditional diffusion for controllable traffic simulation." In 2023 IEEE International Conference on Robotics and Automation (ICRA), pp. 3560-3566. IEEE, 2023.

- This work models the problem as classifier-free guidance incorrectly. This is a classic example of a conditional diffusion, as was done by (Janner et al. 2022) with the initial state being replaced by the history of all agents as well as the map. This is even confirmed by the authors who set the bernoulli p=0.

- Scene-level metrics: the results presented in Table 2 do not seem to be scene-level metrics. Details on the axis of the min should be provided. Scene level metrics (as performed by scene Transformer) measure the minimum average across all agents in a scenes, and not the average minimum across all agents.

- Ablation study results are confusing to me. The results don't seem to support either claim strongly. In addition, it is unclear if the number of parameters on the right side of Figure 5 are equal across all dimensions. Furthermore, why does the minFDE seem to improve as the number of diffusion steps is reduced, except for when the diffusion steps are at 50 baseline (in Figure 5 left)?

- Section 6 does not provide any results, and simply states that it is possible to generate scenes from the long-tail using such an architecture. There are no results to support this claim, as this experiment was performed on an internal dataset. I believe section 6 could/should be placed as part of the conclusion, or the experiment should be performed on a dataset like Argoverse. Also, what is meant by learning the representation of these interactive events? In addition to the results, more details on the method would be necessary, and should be presented in Section 4.

**Questions:**

Questions about the ablation study: The values on the y-axis of Figure 5 seem off compared to Table 3. This is true for both XDE metrics and diversity & multimodality metrics. Can the authors please clarify why that is?

I believe that this work ignores a lot of prior literature that performs multi-agent motion prediction [1, 2] and Transformer specific versions [3, 4], to name just a few.

[1] Gilles, Thomas, Stefano Sabatini, Dzmitry Tsishkou, Bogdan Stanciulescu, and Fabien Moutarde. "Thomas: Trajectory heatmap output with learned multi-agent sampling." arXiv preprint arXiv:2110.06607 (2021).

[2] Rowe, Luke, Martin Ethier, Eli-Henry Dykhne, and Krzysztof Czarnecki. "FJMP: Factorized Joint Multi-Agent Motion Prediction over Learned Directed Acyclic Interaction Graphs." In Proceedings of the IEEE/CVF Conference on Computer Vision and Pattern Recognition, pp. 13745-13755. 2023.

[3] Girgis, Roger, Florian Golemo, Felipe Codevilla, Martin Weiss, Jim Aldon D'Souza, Samira Ebrahimi Kahou, Felix Heide, and Christopher Pal. "Latent variable sequential set transformers for joint multi-agent motion prediction." arXiv preprint arXiv:2104.00563 (2021).

[4] Seff, Ari, Brian Cera, Dian Chen, Mason Ng, Aurick Zhou, Nigamaa Nayakanti, Khaled S. Refaat, Rami Al-Rfou, and Benjamin Sapp. "MotionLM: Multi-Agent Motion Forecasting as Language Modeling." In Proceedings of the IEEE/CVF International Conference on Computer Vision, pp. 8579-8590. 2023.

Details on the implementation would be appreciated. Information like the optimizer used, the inference speed, the number of parameters, etc.

---

### Official Review · Reviewer_imPC · 2023-11-06

**Soundness:** 2 fair
**Presentation:** 3 good
**Contribution:** 2 fair
**Rating:** 3
**Confidence:** 4

**Summary:**

This paper deals with multi-agent trajectory prediction with diffusion models. The authors propose a novel multi-agent trajectory prediction framework, dubbed Scalable Diffusion Transformer (SDT), which is naturally designed to learn the complicated distribution and implicit interactions among agents. SDT is evaluated on a set of real-world benchmark datasets and compared with representative baseline methods, which demonstrates the state-of-the-art multi-agent trajectory prediction ability of SDT in terms of accuracy and diversity.

**Strengths:**

1. The paper is generally well-written and easy to follow.
2. The experimental results seem to support some of the authors' claims.

**Weaknesses:**

1. The authors claimed in the contribution that "it is the first general multi-agent trajectory prediction framework with the diffusion model". However, this is not true. Below are relevant papers that use a diffusion model for multi-agent trajectory prediction. The authors should clearly state what are the major differences between the proposed method and the prior methods, and how these differences lead to meaningful contributions. Moreover, these diffusion-based methods should be compared as baselines to demonstrate the effectiveness of the proposed method. Right now, the contributions of this paper seem incremental and not significant enough.
[1] Stochastic Trajectory Prediction via Motion Indeterminacy Diffusion, CVPR 2022.
[2] Leapfrog Diffusion Model for Stochastic Trajectory Prediction, CVPR 2023.

2. The authors claimed that diffusion models have advantages in complex distribution learning. However, I did not see any visualization of the learned distribution in the qualitative results. It would be better to visualize the distributions in addition to trajectories.

**Questions:**

Please focus on addressing the issues in the Weaknesses section.